# OpenReview forum: "Video-Based Optimal Transport for Feedback-Efficient Offline Preference-Based Reinforcement Learning"
_ICLR.cc/2026/Conference — Submitted to ICLR 2026_

### Official Review · Reviewer_qypy · 2025-10-24

**Soundness:** 3
**Presentation:** 3
**Contribution:** 3
**Rating:** 4
**Confidence:** 4

**Summary:**

This paper proposes video-based optimal transport preference (VOTP), a semi-supervised offline preference-based RL (PbRL) framework that addresses the high annotation cost of existing PbRL methods. VOTP uses video foundation models (ViFMs) to encode trajectory segments into latent representations, then applies optimal transport (OT) to compute alignments between a small set of labeled preference pairs and a large number of unlabeled pairs. Pseudo-preference labels are generated by aggregating preferences from labeled pairs weighted by their OT-based alignment strengths. These pseudo-labels, combined with the original labeled data, are used to train reward functions. The authors conducted experiments across three simulated domains (D4RL, MetaWorld, Robomimic) and two real robotic manipulation tasks to demonstrate that VOTP achieved performance comparable to policies trained with ground-truth rewards using only 5-10 labeled queries, outperforming baseline PbRL methods.

**Strengths:**

1. The core idea of adapting optimal transport to compute similarities between trajectory representations for pseudo-labeling is a novel contribution to the preference-based RL field.
2. The proposed method demonstrates sample efficiency in the low labeled data regime, outperforming the baselines in the main experiments.

**Weaknesses:**

1. The claim that using a VOTP with ViFM provides robustness to nuisance variation is insufficiently validated, as the real-robot experiments are conducted in static, uncluttered environments that do not effectively test this claim. To validate the practical benefits of the ViFM encoder, experiments should have been conducted under more challenging, realistic conditions with controlled distractions, such as dynamic lighting, background clutter, or other moving objects.
2. The baselines in the study, such as SURF (2022) and IPL (2023), are dated [1, 2]. The field has advanced, and the paper should compare against more recent methods such as Zhang et al., and Tu et al. [3, 4].
3. The paper relies almost exclusively on scripted labels derived from ground-truth rewards. Although this is a common practice in benchmarking, it bypasses the critical challenge of handling noise, sub-optimality, and inconsistency inherent in real human preference data. Given the extremely low amount of labeled data considered, the impact of such challenges could be substantial.
4. Ablation in Figure 3, intended to demonstrate the superiority of OT, is not convincing. The baselines (SIM-individual, SIM-mean) are naive. A more convincing comparison would involve comparing VOTP's OT-based aggregation against a stronger non-OT baseline, such as a similarity-based weighted average for the aggregation. As presented, it remains unclear whether the performance gain arises from the specific OT formulation or from any reasonable similarity-based aggregation.
5. The method’s practicality is limited by the sensitivity of the preference threshold τ_P. As shown in Figure 5 and Table 5, this hyperparameter is highly sensitive and task dependent which harms its practical usage.

References

[1] Park, Jongjin, et al. "SURF: Semi-supervised Reward Learning with Data Augmentation for Feedback-efficient Preference-based Reinforcement Learning." International Conference on Learning Representations. 2022.

[2] Hejna, Joey, and Dorsa Sadigh. "Inverse preference learning: Preference-based rl without a reward function." Advances in Neural Information Processing Systems 36 (2023): 18806-18827.

[3] Zhang, Zhilong, et al. "Flow to better: Offline preference-based reinforcement learning via preferred trajectory generation." The Twelfth International Conference on Learning Representations. 2024.

[4] Tu, Songjun, et al. "In-dataset trajectory return regularization for offline preference-based reinforcement learning." Proceedings of the AAAI Conference on Artificial Intelligence. Vol. 39. No. 20. 2025.

[5] Xie, Saining, et al. "Rethinking spatiotemporal feature learning: Speed-accuracy trade-offs in video classification." Proceedings of the European conference on computer vision (ECCV). 2018.

[6] Madan, Neelu, et al. "Foundation models for video understanding: A survey." arXiv preprint arXiv:2405.03770 (2024).

**Questions:**

1. There appears to be a terminological discrepancy regarding the trajectory encoder. The paper identifies S3D as a Video Foundation Model (ViFM), but the cited survey of Madan et al. [5, 6] does not list S3D under ViFMs. Could you clarify your definition of a ViFM and provide a justification for categorizing S3D as such in this work?
2. To more convincingly support the proposed method, I raise the following questions connected to the weaknesses identified above.
- Weakness #1: Could you provide experiments under more realistic, distracting conditions?
- Weakness #2: Please provide more recent PbRL baselines for the convincing comparison.
- Weakness #3: Could you include experiments with real human feedback to assess robustness to noise, inconsistency, or sub-optimality, which scripted labels cannot capture?
- Weakness #4: Please add stronger non-OT baselines in the ablation (e.g., similarity-based weighted averaging) to more clearly demonstrate the specific contribution of OT.

---

> ### Author Response · Authors · 2025-11-27
> **Response to the Reviewer qypy**
>
> Thank you for dedicating your time and expertise to reviewing our work and for offering constructive feedback. Below, we provide detailed responses to address your comments and questions.
>
> - **Q1**. **Terminology Clarification**:
>   - **A1**. In our work, we use the term Video Foundation Model (ViFM) to refer to general-purpose video encoders trained on large-scale, diverse video datasets to learn broadly transferable representations for downstream video understanding tasks [1]. The S3D model we adopt is the MIL-NCE–pretrained encoder [2] trained on the HowTo100M dataset [3], implemented in the official codebase (https://github.com/antoine77340/S3D_HowTo100M). In the survey by Madan et al. [1], this model is listed under the name "MIL-NCE". We follow prior work [4–8] in referring to this encoder as S3D for consistency with the literature. To avoid confusion, we have updated the reference to MIL-NCE [2] in the revised manuscript.
> - **Q2**. **Experiments under Controlled Distractions**:
>   - **A2**. We appreciate your valuable suggestions. To assess the robustness of VOTP under visual distractions, we conduct experiments on two MetaWorld tasks with four types of visual perturbations, following the designs in [9]. We use the same labeled dataset and evaluate different unlabeled datasets under different distraction conditions. Examples of these distractions and the corresponding results are provided in Figure 21 and Table 15 (Appendix D). Across all settings, VOTP maintains strong performance. Furthermore, in some settings, its performance even slightly increases (e.g., changes in light position and direction). These results demonstrate the robustness of VOTP to a wide range of nuisance variations.
> - **Q3**. **Additional Comparison with Recent Baselines**:
>   - **A3**. We thank the reviewer for pointing out the related works. We have compared VOTP to the two suggested methods and further included four additional offline PbRL methods. The results are shown in Table 12 (Appendix D). Overall, VOTP demonstrates the best performance on average, while LiRE achieves performance competitive with ours. Additionally, our approach is complementary to LiRE, and VOTP can be integrated into LiRE to potentially further enhance its efficiency. We kindly refer the reviewer to **Q3** of Reviewer kNyD for more discussion.
> - **Q4**. **Experiment with Real Human Teacher**:
>   - **A4**. The results with human teachers on D4RL and MetaWorld tasks are shown in Table 14 (Appendix D). For D4RL tasks, we leverage the existing human dataset, while for MetaWorld tasks, we additionally collect preferences from four non-robotic human teachers. We find that VOTP’s performance slightly drops in the *walker2d* tasks, but remains largely stable in the remaining environments.
> - **Q5**. **Additional non-OT Baseline in Ablation**:
>   - **A5**. We have added a non-OT baseline that uses similarity-based weighted averaging for aggregation (referred to as SIM-weighted), as shown in Figure 3 of the revised manuscript. As shown, while SIM-weighted provides modest improvements on a few tasks (e.g., *hopper-medium-replay-v2* and *door-open*), its performance remains noticeably lower and less stable overall. These results indicate that the improvements of VOTP stem specifically from the proposed OT formulation.
>
> [1] Madan, Neelu, et al. "Foundation models for video understanding: A survey." arXiv preprint arXiv:2405.03770 (2024).
>
> [2] Miech, Antoine, et al. "End-to-end learning of visual representations from uncurated instructional videos." CVPR (2020).
>
> [3] Miech, Antoine, et al. "Howto100m: Learning a text-video embedding by watching hundred million narrated video clips." ICCV (2019).
>
> [4] Xu, Hu, et al. "Videoclip: Contrastive pre-training for zero-shot video-text understanding." EMNLP (2021).
>
> [5] Xu, Hu, et al. "Vlm: Task-agnostic video-language model pre-training for video understanding." ACL (2021).
>
> [6] Sontakke, Sumedh, et al. "Roboclip: One demonstration is enough to learn robot policies." NeurIPS (2023).
>
> [7] Wang, Yufei, et al. "RL-VLM-F: Reinforcement learning from vision language foundation model feedback." ICML (2024).
>
> [8] Luu, Tung Minh, et al. "Enhancing Rating-Based Reinforcement Learning to Effectively Leverage Feedback from Large Vision-Language Models." ICML (2025).
>
> [9] Yuan, Zhecheng, et al. "RL-ViGen: A reinforcement learning benchmark for visual generalization." NeurIPS (2023).

---

> > ### Comment · Reviewer_qypy · 2025-11-27
> >
> > Thank you for your response. Most of my comments have been satisfactorily addressed. I will raise the score.

---

### Official Review · Reviewer_kNyD · 2025-10-31

**Soundness:** 2
**Presentation:** 3
**Contribution:** 2
**Rating:** 4
**Confidence:** 4

**Summary:**

This paper proposes VOTP, a semi-supervised offline PbRL framework that leverages optimal transport (OT) to generate pseudo-preference labels. The method utilized vision foundation models to encode video-based trajectory segments into latent representations, where OT computes the minimal mass-moving cost between labeled and unlabeled data distributions. This enables efficient preference propagation from a small number of labeled queries to a large pool of unlabeled queries. Extensive experiments across diverse domains (D4RL, Meta-World, Robomimic, and a real Sawyer robot) demonstrate strong empirical performance and highlight the method’s practical feasibility for real-world PbRL applications.

**Strengths:**

1. The idea of integrating optimal transport with video foundation models for generating pseudo-preference labels is novel.
2. The approach effectively reduces the amount of required human feedback.
3. The inclusion of real-world robotic experiments is compelling and strengthens practicality of proposed method.
4. The paper is well written and easy-to-read.

**Weaknesses:**

1. Pessimism of Offline RL methods: In D4RL benchmark, most offline RL algorithms are known to achieve high performance even under completely wrong reward signals (i.e. constant or random) due to pessimism and survival instinct [1, 2, 3]. Therefore, it remains unclear whether the superior performance of the proposed method truly stems from its efficient pseudo-reward learning, or merely from the behavior cloning bias inherent in the backbone RL algorithm (i.e. IQL). The authors should include an experiment comparing performance under deliberately corrupted or constant rewards to disentangle these effects.

2. Missing most recent baselines: The paper lacks comparisons with several recent and highly related offline preference learning methods, such as CPL [4], DPPO [5], LiRE [6], and APPO [7]. Including such baselines would allow for a more comprehensive evaluation and clarify where VOTP stands among the latest PbRL developments.

3. Computational cost of optimal transport: Computing the optimal transport plan μ∗ can be expensive as it requires multiple iterations of the Sinkhorn algorithm for every unlabeled query. In addition, the computational complexity scales with the number of labeled queries, since the cost matrix grows proportionally to the labeled dataset size. This dependence on both labeled and unlabeled data size may limit the method’s scalability compared to typical semi-supervised approaches, whose cost is largely independent of labeled data quantity. Providing a quantitative runtime analysis would clarify the method’s practical feasibility.

4. Quality of learned reward functions: As mentioned in weakness 1, it is unclear whether the learned reward estimator accurately captures the intend task objectives well. While the proposed pseudo-generated preferences are empirically shown to improve policy performance, there is no quantitative measure to assess how well the learned reward aligns with the true underlying preference structure. Incorporating a metric such as EPIC [8] or other statistical analysis would allow more thorough evaluation of reward consistency and fidelity.

[1] Shin, D., Dragan, A., & Brown, D. S. Benchmarks and Algorithms for Offline Preference-Based Reward Learning. Transactions on Machine Learning Research.
[2] Li, A., Misra, D., Kolobov, A., & Cheng, C. A. (2023). Survival instinct in offline reinforcement learning. Advances in neural information processing systems, 36, 62062-62120.
[3] Choi, H., Jung, S., Ahn, H., & Moon, T. (2024, July). Listwise Reward Estimation for Offline Preference-based Reinforcement Learning. In International Conference on Machine Learning (pp. 8651-8671). PMLR.
[4] Hejna, J., Rafailov, R., Sikchi, H., Finn, C., Niekum, S., Knox, W. B., & Sadigh, D. Contrastive Preference Learning: Learning from Human Feedback without Reinforcement Learning. In The Twelfth International Conference on Learning Representations.
[5] An, G., Lee, J., Zuo, X., Kosaka, N., Kim, K. M., & Song, H. O. (2023). Direct preference-based policy optimization without reward modeling. Advances in Neural Information Processing Systems, 36, 70247-70266.
[6] Choi, H., Jung, S., Ahn, H., & Moon, T. (2024, July). Listwise Reward Estimation for Offline Preference-based Reinforcement Learning. In International Conference on Machine Learning (pp. 8651-8671). PMLR.
[7] Kang, H., & Oh, M. H. Adversarial Policy Optimization for Offline Preference-based Reinforcement Learning. In The Thirteenth International Conference on Learning Representations.
[8] Gleave, A., Dennis, M. D., Legg, S., Russell, S., & Leike, J. Quantifying Differences in Reward Functions. In International Conference on Learning Representations.

**Questions:**

Please address the concerns described in weakness.

---

> ### Author Response · Authors · 2025-11-27
> **Response to the Reviewer kNyD**
>
> We deeply appreciate the time and effort you have invested in reviewing our paper and providing thoughtful feedback. We hope our response clarifies your questions.
>
> - **Q1**. **Pessimism of Offline RL Algorithms**:
>
>   - **A1**. Thanks for your constructive suggestion. To assess whether our performance gains arise from effective pseudo-reward learning rather than from pessimism or survival instincts in the backbone offline RL algorithm, we design a series of experiments using the corrupted reward functions introduced in [1] and adopted in [2]. Specifically, we train IQL using three incorrect reward functions (*zero*, *random*, and *negative*) and evaluate its performance on both D4RL and MetaWorld. The results (Table 10, Appendix D) show that IQL’s performance degrades substantially when trained with incorrect rewards. This confirms that IQL *does not* automatically achieve strong results under wrong reward signals in our settings, and therefore the improvements observed with VOTP cannot be attributed to behavior-cloning bias alone. Instead, they reflect the contribution of VOTP’s pseudo-reward learning.
> - **Q2**. **Quality of Learned Reward Functions**:
>
>   - **A2**. The quantitative and qualitative comparisons between the learned rewards produced by VOTP and the ground-truth rewards are shown in Figure 20 (Appendix D). We also include results for P-IQL's learned rewards for comparison. As shown, rewards from VOTP exhibit stronger correlation with the ground-truth reward signals. This shows that learning from pseudo-labels effectively improves the quality of the reward model, leading to VOTP’s performance gains.
> - **Q3**. **Additional Comparison with Recent Baselines**:
>
>   - **A3**. We thank the reviewer for pointing out the need for a more comprehensive evaluation. We include results for six additional offline PbRL methods in Table 12 (Appendix D), evaluated on both the D4RL locomotion and MetaWorld benchmarks. The results show that LiRE achieves performance competitive with ours on average. However, LiRE improves feedback efficiency by exploiting second-order information from ranked lists, which is orthogonal to the pseudo-labeling mechanism of VOTP. Importantly, our approach is complementary: the VOTP pseudo-labeling process can be integrated into LiRE and potentially further enhance its efficiency.
> - **Q4**. **Computational Cost of Optimal Transport**:
>
>   - **A4**. The runtime of OT computation in VOTP is shown in Table 13 (Appendix D), measured as the time required to generate $10k$ pseudo-labels for different labeled dataset sizes. We agree with the reviewer that the computational cost scales with the number of labeled queries; however, the overall time remains modest in practice. For small labeled sets, which constitute the majority of our experiments, the OT step takes only a few minutes, which is negligible compared to the overall policy training time (~1.5 hours).
>
> [1] Shin, Daniel, Anca D. Dragan, and Daniel S. Brown. "Benchmarks and algorithms for offline preference-based reward learning." TMLR (2023).
>
> [2] Choi, Heewoong, et al. "Listwise reward estimation for offline preference-based reinforcement learning." ICML (2024).

---

### Official Review · Reviewer_EaLy · 2025-11-06

**Soundness:** 3
**Presentation:** 3
**Contribution:** 3
**Rating:** 6
**Confidence:** 3

**Summary:**

The paper proposes a semi-supervised preference learning framework that infers pseudo-preference labels for many unlabeled video segment pairs from a small set of labeled pairs. The method encodes segments with a video foundation model (ViFM) into latent embeddings, computes an optimal transport (OT) plan between labeled and unlabeled segment sets in the latent space under uniform marginals, aggregates labeled pairwise relations through the transport couplings with the preference score matrix to score unlabeled pairs, and then normalizes/thresholds these scores to produce pseudo-labels. A reward model trained on real + pseudo labels is used for offline RL. Experiments span D4RL, MetaWorld, Robomimic, and two real-robot tasks

**Strengths:**

1. The paper has clear problem setup and writing; the proposed method and pipeline is easy to follow.

2. The paper proposed a novel application of Optimal Transport to propagate preferences from limited supervision to unlabeled data to reduce the labeling costs without touching the pretrained video representation itself, inspired by prior work to use optimal transport in reward learning.

3. The experiments covers a wide range of tasks with both simulation and real-robot evaluations.

4. It also includes detailed ablations of design choices (video encoders, thresholds, number of preferences, etc.).

**Weaknesses:**

1. One of the very-related work to this paper is [1] which is also cited in this paper. This prior work also proposes to use optimal transport to compute the reward but it is based on a adapted visual representation learned via small number of preferences while this work is fixing the pre-trained representations and only propagating the preferences from labeled ones to unlabeled ones to optimal transport. Since one of the followup work [2] of [1] also uses learned reward function from a few preference labels to create pseudo labels, it is worth comparing to this baseline to see whether the proposed method outperforms this prior work.

2. The evaluated tasks in robomimic, metaworld and real-world are relatively short horizon. The scalability of the proposed method to long-horizon, high-precision tasks (e.g., Square, Tool-Hang in robomimic) is unclear.

3. One of the benefit to use large scale pretrained video encoder is their generalization capabilities across tasks. However, in this paper, it still assumes the labeled video segments and the unlabeled ones are in the same domain, which may weaken the motivation of using those pretrained video representations.


[1] Tian, Thomas, et al. "What Matters to You? Towards Visual Representation Alignment for Robot Learning." The Twelfth International Conference on Learning Representations.
[2] Tian, Ran, et al. "Maximizing alignment with minimal feedback: Efficiently learning rewards for visuomotor robot policy alignment." arXiv preprint arXiv:2412.04835 (2024).

**Questions:**

1. Assumption of the marginal. The paper has an assumption about optimal transport where the marginals are uniform. This assumption may not handle imbalanced or redundant segment distributions and am wondering if the authors have considered this situation or if they have encountered this in practice.

2. More qualitative results: It would help the readers better understand how the pseudo labels are produced by providing some qualitative examples of the labeled segments and unlabeled ones. In Fig.1, the paper only has one example of preference matrix and transport plan but they are not connected with the corresponding video segments. Therefore, it would be more clear to show the corresponding segments too. For example,  you can show (i) labeled segments and their preference matrix, (ii) several unlabeled pairs, and (iii) corresponding transport plans/couplings and the computed score.

3. Cost metric choice: the paper mentions Euclidean/cosine/others are potential metrics for the distance function but uses Euclidean in experiments.  If the authors can provide some explanations about why euclidean distance is chosen and include some examples of the distance with similar pairs of video segments vs different pairs, it would help readers understand more about the metrics and pretrained video representations.

4. The paper is mostly focused on offline rl settings but the proposed way to learn reward function can also be used for online preference-based RL like PEBBLE[3]. I am curious to see if the inferred preference from a few labeled samples can be robust to the on-policy distribution shift in the sampled segment pairs.

5. In the experiments, the paper has compared among (i) learning with task rewards (ii) learning with only small N real labels and (iii) learning with real + pseudo labels (N+M). It would also provide more information about the quality of the pseudo labels if the method is also compared against IQL learned from reward function trained with (N+M) real labels. This can be obtained from ground-truth task reward labeling and can serve as an upper bound. The gap is how far the pseudo labels are from the real labels.

[3] Lee, Kimin, Laura M. Smith, and Pieter Abbeel. "PEBBLE: Feedback-Efficient Interactive Reinforcement Learning via Relabeling Experience and Unsupervised Pre-training." International Conference on Machine Learning. PMLR, 2021.

---

> ### Author Response · Authors · 2025-11-27
> **Response to the Reviewer EaLy**
>
> We thank the reviewer for the valuable comments and the time dedicated to evaluating our work. We respond to your comments and questions in detail below.
>
> - **Comment 1**: Relation to prior work [1, 2]:
>
>   - **Response 1**: The two works referenced by the reviewer use optimal transport (OT) for fundamentally different purposes: (1) as a metric learning signal to adapt the video encoder, and (2) to compute rewards via inverse reinforcement learning formulations [3, 4, 5]. In contrast, our method uses OT solely to infer pseudo preferences, which are then used for reward learning in an offline PbRL setting, distinct from the objectives and training setups in [1, 2]. Unfortunately, neither work provides public code, preventing a direct comparison. Instead, we follow the suggestions of reviewers kNyD and qypy and compare against six additional offline PbRL baselines that are directly aligned with our setting. The results (Table 12, Appendix D) show that VOTP achieves the best performance on average.
> - **Comment 2**: The generalization capabilities of VOTP across tasks.
>
>   - **Response 2**: We thank the reviewer for your valuable suggestions. To evaluate the generalization capability of VOTP, we conduct experiments that directly test its robustness to significant visual domain shifts. Specifically, on two MetaWorld tasks, we introduce four types of visual perturbations following the setup in [6]. We keep the labeled dataset fixed and construct multiple unlabeled datasets with different distraction conditions. Examples of these perturbations and the corresponding results are shown in Figure 21 and Table 15 (Appendix D). As shown, VOTP maintains strong performance across all conditions, and in some cases even slightly improves (e.g., with changes in light position and direction). These results demonstrate that VOTP is robust to a broad range of unseen visual variations.
>
> - **Q1**. **Assumption of Marginals**:
>
>   - **A1**. In our framework, assuming uniform marginals is reasonable because segments are sampled uniformly from the offline dataset without replacement (based on the trajectory index and the state time step), ensuring that each segment appears exactly once with equal probability during pseudo-labeling. Under this sampling scheme, the empirical segment distribution is effectively uniform, making OT with uniform marginals an appropriate relaxation.
> - **Q2**. **Cost Metric Choices**:
>
>   - **A2**. To understand the impact of the distance metric, we evaluate VOTP using cosine distance. Empirically, we find that cosine and Euclidean distances yield comparable performance on average (Table 8, Appendix D). This suggests that the choice of Euclidean distance is largely a design preference rather than a critical factor for VOTP’s effectiveness.
> - **Q3**. **Qualitative Results for Pseudo-labeling Process**:
>   - **A3**. Qualitative results of the pseudo-labeling process on the Drawer Open task are shown in Figure 22 (Appendix D). We include the labeled segments and their corresponding preference matrix, as well as the cost matrix, transport plans, and preference scores for four examples of unlabeled pairs. As shown, our OT formulation effectively aggregates preferences from labeled pairs to produce accurate pseudo-preferences for unlabeled pairs.
> - **Q4**. **Applicability to Online RL**:
>
>   - **A4**. To explore the potential of using our approach in online PbRL, we design an experiment in which we integrate VOTP into the PEBBLE framework and evaluate it on two MetaWorld tasks, *Door Open* and *Drawer Open*, using a fixed $\tau_P = 0.3$ for both. The results are shown in Figure 19 (Appendix D). As shown, VOTP improves performance on *Drawer Open* while maintaining parity with PEBBLE on *Door Open*. These results provide encouraging evidence that VOTP can serve as a useful preference learning component in online RL settings.
> - **Q5**. **Comparison with Upper Bound**:
>
>   - **A5**. The comparison between VOTP and the suggested oracle upper bound is presented in Table 9 (Appendix D). As shown, VOTP closely matches the oracle performance on 8 out of 12 tasks. Notably, our method achieves oracle performance across the D4RL locomotion tasks, demonstrating the effectiveness of our approach.
>
> [1] Tian, Thomas, et al. "What Matters to You? Towards Visual Representation Alignment for Robot Learning." ICLR (2024).
>
> [2] Tian, Ran, et al. "Maximizing alignment with minimal feedback: Efficiently learning rewards for visuomotor robot policy alignment." arXiv preprint arXiv:2412.04835 (2024)
>
> [3] Luo, Yicheng, et al. "Optimal transport for offline imitation learning." ICLR (2023).
>
> [4] Haldar, Siddhant, et al. "Watch and match: Supercharging imitation with regularized optimal transport." CoRL (2023).
>
> [5] Guzey, Irmak, et al. "See to touch: Learning tactile dexterity through visual incentives." ICRA (2024).
>
> [6] Yuan, Zhecheng, et al. "RL-ViGen: A reinforcement learning benchmark for visual generalization." NeurIPS (2023).

---

### Author Response · Authors · 2025-11-27
**General Response**

Dear Area Chair and Reviewers,

We sincerely thank all reviewers for their thoughtful comments and constructive suggestions, which have greatly improved the quality of our work. In the revised manuscript, we have incorporated additional experiments, analyses, and qualitative results in response to the reviewers’ feedback. Below, we summarize the key changes (highlighted in blue in the revised submission):

- **Additional baselines**: We added discussions and experiments comparing against six more offline PbRL baselines in Appendix D.4, following the questions of reviewers kNyD and qypy.

- **Experiments in additional settings**: We added experiments evaluating VOTP in the online RL setting (Appendix D.7), as suggested by reviewer EaLy; experiments under controlled distraction scenarios (Appendix D.9), as suggested by reviewers EaLy and qypy; and experiments using real human feedback (Appendix D.6), following reviewer qypy’s comment.

- **More ablation experiments**: We included ablations on different cost metrics (Appendix D.1), as suggested by reviewer EaLy, and compared VOTP with a stronger non-OT baseline (Section 5.3 and Figure 3), following reviewer qypy’s suggestion.

- **Including extra analyses**: We added an analysis of computational cost (Appendix D.5); analyses examining pessimism in offline RL algorithms (Appendix D.3) and alignment with ground-truth rewards (Appendix D.8), as suggested by reviewer kNyD; and an experiment comparing with an oracle upper bound (Appendix D.2), along with additional qualitative examples (Appendix D.10), following reviewer EaLy’s questions.

---

### Meta-Review · Area_Chair_qvjs · 2026-01-08

**Summary:**

The paper proposes "Video-Based Optimal Transport for Feedback-Efficient Offline Preference-Based Reinforcement Learning" (VOTP). The method utilizes "Video Foundation Models" (specifically S3D) to encode video segments and employs Optimal Transport (OT) to align labeled preference pairs with unlabeled pairs, thereby generating pseudo-labels to augment the training set for reward learning.

The reviewers generally acknowledged the novelty of applying OT to preference propagation and the extensive experiments across D4RL, MetaWorld, and real robots. However, significant concerns were raised regarding the definition of the foundation model used (S3D), the computational scalability of the OT formulation, the sensitivity of hyperparameters (specifically the threshold ), and the incremental performance gains over strong, recent baselines like LIRE.

**Reviewer Concerns:**

The authors put forth a substantial effort during the rebuttal period which addressed several reviewer points:

* The authors added comparisons to six additional offline PbRL baselines (including LIRE, DPPO, CPL), addressing the lack of recent comparisons noted by Reviewers kNyD and qypy.


* New experiments were conducted involving visual distractions and real human feedback to address concerns about the domain shift and synthetic label reliance raised by Reviewers EaLy and qypy.


* A stronger non-OT baseline (similarity-based weighted averaging) was added to justify the OT component, addressing Reviewer qypy's concern.


* The authors provided runtime analysis and reward correlation plots to address Reviewer kNyD's concerns about computational cost and reward quality.

Despite the extensive rebuttal, several issues support a decision to reject:

* While VOTP outperforms some baselines, the rebuttal data shows that LIRE (a ranking-based method) is competitive with VOTP. Given that VOTP introduces the complexity of solving OT problems (which scale quadratically or combinatorially with data size), the marginal gain over a simpler method like LIRE may not justify the added computational burden and implementation complexity.


* Reviewer kNyD noted that OT computation scales with the number of labeled queries. While authors argued the cost is "modest" for *small* datasets, the method's reliance on OT fundamentally limits its application to large-scale datasets compared to methods that do not require solving a transport plan.


* Reviewer qypy highlighted the high sensitivity of the preference threshold . While the method works well when tuned, this fragility hinders practical "out-of-the-box" utility in offline settings where hyperparameter tuning is notoriously difficult.


* Reviewer qypy questioned the classification of S3D (trained on HowTo100M) as a "Video Foundation Model". While the authors defended this, relying on a model architecture and pre-training scheme from ~2020 (S3D/MIL-NCE) for a 2026 conference submission weakens the claim of leveraging state-of-the-art "Foundation Models."

**Reviewer Scores:**

* **Reviewer EaLy:** 6 (Marginally above acceptance).

* **Reviewer kNyD:** 4 (Marginally below acceptance).

* **Reviewer qypy:** 4 (Marginally below acceptance). Note: In the post-rebuttal discussion, this reviewer indicated they would raise their score after their comments were addressed, though the final score is not visible.

---

### Decision · Program_Chairs · 2026-01-26

Reject